# Nanomechanical and Thermodynamic Alterations of Red Blood Cells in Chronic Lymphocytic Leukemia: Implications for Disease and Treatment Monitoring

**DOI:** 10.3390/ijms27010353

**Published:** 2025-12-29

**Authors:** Velichka Strijkova, Vesela Katrova, Miroslava Ivanova, Ariana Langari, Lidia Gartcheva, Margarita Guenova, Anika Alexandrova-Watanabe, Stefka G. Taneva, Sashka Krumova, Svetla Todinova

**Affiliations:** 1Institute of Optical Materials and Technologies “Acad. Yordan Malinovski”, Bulgarian Academy of Sciences, “Acad. G. Bontchev” Str. 109, 1113 Sofia, Bulgaria; vily@iomt.bas.bg (V.S.); vlozanova@iomt.bas.bg (V.K.); 2Institute of Biophysics and Biomedical Engineering, Bulgarian Academy of Sciences, “Acad. G. Bontchev” Str. 21, 1113 Sofia, Bulgaria; miroslava.ilieva.ivanova@gmail.com (M.I.); arianalangari@abv.bg (A.L.); sgtaneva@gmail.com (S.G.T.); sashka.b.krumova@gmail.com (S.K.); 3Center of Competence at Mechatronics and Clean Technologies—MIRACle, “Acad. G. Bontchev” Str. 4, 1113 Sofia, Bulgaria; anikaalexandrova@abv.bg; 4National Specialized Hospital for Active Treatment of Hematological Diseases, Zdrave Str. 2, 1756 Sofia, Bulgaria; l.garcheva@hematology.bg (L.G.); m.genova@hematology.bg (M.G.); 5Institute of Mechanics, Bulgarian Academy of Sciences, “Acad. G. Bontchev” Str. 4, 1113 Sofia, Bulgaria

**Keywords:** chronic lymphocytic leukemia, obinutuzumab/venetoclax, ibrutinib, red blood cells, atomic force microscopy, differential scanning calorimetry

## Abstract

Chronic lymphocytic leukemia (CLL) has systemic effects that extend beyond malignant lymphocytes, potentially altering the structure and function of circulating red blood cells (RBCs). In this study, atomic force microscopy (AFM) was combined with complementary calorimetric analysis to investigate the membrane ultrastructure, nanomechanical characteristics, and thermodynamic behavior of RBCs from untreated CLL patients and those receiving targeted therapies (Obinutuzumab/Venetoclax or Ibrutinib). RBCs from untreated patients exhibited pronounced reduction in membrane roughness, increased stiffness and adhesion forces, and altered thermal unfolding of cytoskeletal and membrane proteins, indicative of impaired structural flexibility and stability. Treatment with Obinutuzumab/Venetoclax partially restored surface topography, but stiffness and adhesion forces remained elevated, suggesting persistent cytoskeletal rigidity. The obscured spectrin and Band 2–4 thermal transitions and the elevated total enthalpy change revealed by differential scanning calorimetry indicated a modified conformation or binding state of membrane proteins. In contrast, Ibrutinib therapy produced near-normal nanomechanical and thermal characteristics, reflecting a more comprehensive restoration of RBC integrity. These findings demonstrate that CLL and its therapies distinctly influence erythrocyte morphology and mechanics, underscoring the systemic impact of the disease. The strong correspondence between AFM and calorimetric data highlights the potential of integrated biophysical approaches to detect subtle RBC alterations and to serve as complementary indicators for therapeutic monitoring.

## 1. Introduction

Chronic lymphocytic leukemia (CLL), the most common leukemia in adults, is characterized by the clonal accumulation of dysfunctional B-lymphocytes in the bone marrow, blood, and lymphoid tissues. CLL is a highly heterogeneous disease influenced by multiple factors [1]. The status of the immunoglobulin heavy chain variable region (IGHV) is a key determinant of disease behavior. An unmutated IGHV is associated with a more aggressive clinical course and poorer response to chemoimmunotherapy [2]. Despite its diminished prognostic significance in the context of targeted therapies, this marker remains clinically relevant [3]. High expression of CD38 and ZAP-70, as well as the presence of cytogenetic abnormalities, such as del(13q14), trisomy 12, del(11)(q22.3), and del(17p13), are indicative of unfavorable outcomes [4,5]. Together, these parameters can be used to classify CLL as either an indolent or a rapidly progressive disease.

Advances in the understanding of biological heterogeneity in CLL have transformed treatment from chemoimmunotherapy to targeted therapies [6]. Venetoclax, a selective BCL-2 inhibitor, induces apoptosis through BAX/BAK activation and demonstrates high efficacy in both treatment-naïve and relapsed/refractory CLL [7,8]. BTK inhibitors represent another major class of therapeutic agents, particularly for high-risk CLL patients harboring del(17p), TP53 mutations, or unmutated IGHV. Ibrutinib is the first covalent BTK inhibitor. It blocks the signaling of the B-cell receptor (BCR), thereby impairing malignant B cells. However, its use is associated with adverse events, including bleeding, atrial fibrillation, and hypertension [9,10,11].

While targeted therapies such as Venetoclax and BTK inhibitors have significantly improved the management of CLL, the disease exerts systemic effects that extend beyond leukemic cell burden. There is increasing evidence that CLL affects other blood components, including red blood cells (RBCs). RBC abnormalities are frequently observed in CLL, with anemia representing one of the most clinically important complications. Autoimmune hemolytic anemia (AIHA) occurs in approximately 4–10% of CLL patients. It reflects immune dysregulation in CLL and is mediated by autoantibodies that prematurely destroy RBCs, reducing their number and functional capacity [12].

In addition to quantitative reductions, leukemia is frequently accompanied by qualitative alterations in RBC morphology and membrane integrity. Majumder et al. reported that in leukemia patients, including those with CLL, RBCs often lose their characteristic biconcave shape and exhibit membrane surface irregularities, increased porosity, and a flaccid appearance [13]. Subsequent work indicated that such morphological alterations arise from disruptions in RBC membrane organization, which play a critical role in regulating cell deformability in pathological conditions, including anemias [14,15].

Recent studies, using microfluidic and imaging flow analysis, have provided evidence of these structural alterations in CLL. RBCs from untreated CLL patients demonstrated abnormal aggregation and the formation of large, stable three-dimensional (3D) clusters and exhibited impaired shear-induced disaggregation. These behaviors were not observed in healthy controls. Such rheological alterations indicate changes in the mechanical properties and surface organization of the RBC membrane [16,17].

RBC deformability, the ability of erythrocytes to undergo reversible shape changes in response to mechanical forces, is a critical determinant of microvascular perfusion and oxygen transport. This property depends on the structural integrity of the RBC membrane, which comprises a lipid bilayer supported by the cytoskeletal network. Membrane disruptions are observed in numerous hematological and systemic disorders and correlate with disease severity and related complications [18,19,20].

Taken together, these observations suggest that RBCs in CLL and other leukemias undergo qualitative changes that warrant further investigation. Such abnormalities may compromise microvascular perfusion, reduce oxygen transport efficiency, and disrupt microcirculatory flow, potentially contributing to tissue hypoxia, disease-related fatigue, and organ dysfunction. Collectively, they highlight the importance of complementary biophysical approaches to elucidate RBC membrane alterations in leukemia.

Despite its clinical relevance, research into the nanomechanical properties of RBCs in CLL remains limited. Conventional hematological analyses primarily focus on leukemic cell populations and overall blood counts, providing only limited insight into RBC mechanical behavior. Moreover, the impact of targeted therapies, such as BTK inhibitors and anti-CD20 monoclonal antibodies in combination with BCL-2 inhibitors, on RBC morphology and nanomechanics has not been comprehensively investigated. Clarifying these changes is important, as alterations in RBC biophysical properties can affect patient symptoms and clinical outcomes. In this context, atomic force microscopy (AFM) provides a powerful tool for nanoscale imaging and mechanical probing, enabling quantitative assessment of RBC membrane topography and stiffness. Differential scanning calorimetry (DSC) offers complementary information on the stability of hemoglobin and the main components of the cytoskeleton and plasma membrane in intact cells, which can be affected in disease. Therefore, combining AFM with optical microscopy, DSC, and microfluidic assays enables multifaceted characterization of erythrocyte functionality and biophysical properties through complementary approaches.

This study investigates the nanomechanical and thermodynamic properties of RBCs from patients with CLL, with a focus on their relationship to disease-associated alterations and prognostic features.

To this end, AFM and DSC were employed to analyze RBCs from untreated CLL patients and those treated with Ibrutinib or Obinutuzumab/Venetoclax with the following objectives: (i) to identify disease-associated alterations in RBC structure, nanomechanical behavior and thermodynamic stability; (ii) to evaluate the effects of targeted therapies on these parameters; and (iii) to explore the potential relevance of RBC-associated biophysical features for monitoring disease progression and treatment response.

## 2. Results

### 2.1. Clinical and Hematological Characteristics of the CLL Patients and Healthy Individuals

The clinical and hematological characteristics of the study groups are summarized in Table 1. There was a significant difference in mean age among the groups (*p* = 0.04), with patients treated with Obinutuzumab/Venetoclax being older (69.6 ± 7.5 years) than healthy donors (56.6 ± 8.8 years) and untreated CLL patients (60.5 ± 12.3 years). Patients receiving Ibrutinib had an intermediate mean age (64.1 ± 9.2 years), which did not differ significantly from the other groups.

None of the patients had received blood transfusions before sample collection. Overall, most key hematologic parameters did not differ markedly between CLL patients and healthy controls, with mean values largely within their respective reference ranges. Hemoglobin (Hb) concentration was the main exception, being significantly lower in the Obinutuzumab/Venetoclax group than in healthy donors, untreated patients, and those treated with Ibrutinib (*p* = 0.039) (Table 1). Accordingly, mild anemia was observed in a subset of patients receiving Obinutuzumab/Venetoclax.

Significant differences were also observed in RBC counts, with untreated CLL patients and those treated with Ibrutinib showing higher values than healthy donors and patients receiving Obinutuzumab/Venetoclax (*p* = 0.006). Hematocrit tended to be lower in the Obinutuzumab/Venetoclax group (0.41 ± 0.04 L/L), although this difference did not reach statistical significance (*p* = 0.06). Within-group analysis revealed Ht values below the reference range in four patients receiving Obinutuzumab/Venetoclax, compared with one patient in the untreated group. Other erythrocyte indices, including MCV, MCH, MCHC, and RDW, did not differ significantly among the groups. RDW values were within the reference range in all groups, indicating no appreciable variation in erythrocyte size and arguing against increased reticulocytosis. Nucleated red blood cells (NRBCs)/erythroblasts were not detected in the peripheral blood of CLL patients (Table 1).

White blood cell counts and absolute lymphocyte counts were markedly elevated in untreated CLL patients and were significantly higher than in all other groups, consistent with untreated disease status. No significant differences were observed in total bilirubin across the study groups, and platelet counts remained within the reference range in all cases (Table 1).

Comorbidities differed across CLL groups. In the untreated group, one patient had varicose veins, and one had ischemic heart disease. Among patients treated with Obinutuzumab/Venetoclax, arterial hypertension was the most frequent comorbidity (*n* = 5), with additional conditions including atrial flutter (*n* = 1), type 2 diabetes mellitus (*n* = 1), and heart failure (*n* = 2). In the Ibrutinib-treated group, one patient had essential hypertension, and one had a moderately differentiated adenocarcinoma of the sigmoid colon (G2).

### 2.2. RBC Morphology in CLL Patients

To complement standard blood counts, RBC morphology was quantified using optical microscopy. These analyses revealed some morphological heterogeneity across samples that routine blood count measurements did not capture.

RBC shape in samples from healthy donors and from CLL patients, untreated or treated with Obinutuzumab/Venetoclax or Ibrutinib, was assessed using optical microscopy (50×) and AFM. Optical microscopy was used to obtain robust statistics for the quantitative classification of RBC morphologies, as this approach permits the analysis of a larger number of cells than AFM.

Cells were categorized into three classes: biconcave (discocytes), spiculated cells, and spherocytes. Classification followed established criteria: discocytes were defined as normal biconcave cells with central pallor; spiculocytes were characterized by multiple short, evenly distributed membrane projections; and spherocytes were categorized as spherical cells lacking central pallor (Figure 1). The presence or absence of a central pit was used to distinguish among these morphological forms. For each donor, 250–270 cells were analyzed, and the relative proportion of each morphological class was normalized to the total number of counted cells.

Figure 2 shows the relative proportions of these classes obtained by optical microscopy.

Biconcave cells were the dominant type in all groups. However, their proportion was modestly reduced in untreated CLL patients and those receiving Obinutuzumab/Venetoclax, which exhibited higher numbers of spiculated cells and spherocytes. In contrast, the Ibrutinib-treated group exhibited distributions similar to the healthy controls. Although minor differences in the average proportions of morphological types were observed in RBC samples from untreated and Obinutuzumab/Venetoclax-treated patients compared to the other groups, these differences did not reach statistical significance (Figure 2).

Figure 3 shows representative AFM images illustrating group-specific RBC morphological types for healthy controls, untreated CLL patients, and CLL patients receiving Obinutuzumab/Venetoclax or Ibrutinib.

For a more detailed characterization, RBC membrane architecture and elasticity were further assessed at the nanoscale using AFM.

### 2.3. Nanostructural and Nanomechanical Parameters of RBCs in CLL Patients

As part of the AFM analysis, membrane roughness (Rrms), an established indicator of RBC membrane-skeleton integrity [21] and further validated in aging studies [22,23], was used as a key parameter to assess structural alterations of the membrane. High-resolution 3D AFM images depict the nanoscale topography of RBCs from three independent donors/patients per group: healthy donors (panels A_1_–A_3_), untreated CLL patients (panels B_1_–B_3_), patients treated with Obinutuzumab/Venetoclax (panels C_1_–C_3_), and patients treated with ibrutinib (panels D_1_–D_3_) (Figure 4). These images revealed distinct alterations in membrane architecture associated with disease status and therapy response compared with control cells. RBCs from untreated CLL patients exhibited smoother plasma membranes than those of healthy controls. Consistent with this observation, quantitative analysis showed a significant reduction in membrane roughness in untreated CLL RBCs (Rrms = 3.25 ± 0.67 nm vs. 4.79 ± 1.09 nm in controls, *p* < 0.05) (Table 2).

RBCs from patients treated with Obinutuzumab/Venetoclax exhibited partial restoration of the characteristic surface topography, including membrane morphology and roughness (Rrms = 4.98 ± 1.08 nm), comparable to that of healthy controls. RBCs from Ibrutinib-treated patients also displayed near-normal surface features, with a membrane roughness of 4.05 ± 0.92 nm (Figure 4, Table 2).

In addition to evaluating membrane topography, the deformability and surface interactions of RBCs were assessed by measuring Young’s modulus (Ea) and adhesive forces. Force−distance (F−D) curves were employed to quantify the mechanical properties and adhesive behaviors of the RBCs as the AFM tip approached and retracted from the cell membrane.

Analysis of the elastic modulus (Ea), a quantitative measure of membrane stiffness obtained from AFM force–indentation curves (Figure 5), revealed pronounced differences among the studied groups (Table 2). The representative curves shown in Figure 5 illustrate the mechanical response of RBC membranes under increasing load, highlighting group-specific variations in stiffness and deformation behavior. Higher Ea values correspond to increased membrane rigidity and, consequently, reduced deformability.

RBCs from untreated CLL patients exhibited significantly increased stiffness (Ea = 1.48 ± 0.55 MPa) compared with healthy controls (Ea = 0.348 ± 0.11 MPa; *p* < 0.05), indicating pronounced membrane rigidity and reduced cellular deformability. Effects of treatment were also evident, i.e., RBCs from patients receiving Obinutuzumab/Venetoclax exhibited a further increase in Ea (3.30 ± 0.47 MPa), while RBCs from Ibrutinib-treated patients showed a lower Ea (0.595 ± 0.31 MPa), approaching control values (Table 2).

To more accurately characterize the mechanical heterogeneity of the samples, the distribution of Ea values across RBCs was analyzed using histograms (Figure 6). Analysis of these distributions revealed distinct patterns among the groups. RBCs from healthy controls exhibited a narrow distribution, reflecting a consistent mechanical phenotype that is compatible with intact membrane structure and an organized cytoskeleton. In contrast, CLL samples, especially those from untreated and Obinutuzumab/Venetoclax-treated patients, displayed a significantly broader Ea distribution, indicating increased mechanical heterogeneity, likely resulting from membrane reorganization or structural alterations. The distribution of Ea values in Ibrutinib-treated patients was narrower compared with that of the other patient cohorts and more closely resembled that of healthy controls. However, a subset of cells remained outside the normal range, suggesting that membrane mechanical integrity was only partially preserved. These observations are consistent with relatively mild membrane alterations observed in this cohort (Figure 6).

In our measurements, control RBCs exhibited average adhesive forces of approximately 260 pN. Consistent with the trend observed for Young’s modulus, RBCs from untreated CLL patients and those treated with Obinutuzumab/Venetoclax showed significantly higher adhesive force values (~350 pN) reflecting an increase of about 25% (Table 2). These findings indicate a substantial enhancement of membrane-mediated adhesion in CLL.

### 2.4. Calorimetric Properties of RBCs in CLL and the Healthy State

The average thermograms recorded for intact RBCs from healthy donors are presented in Figure 7. In agreement with previously reported calorimetric studies [24,25,26], several well-defined thermal transitions were observed in the DSC profiles of healthy RBCs. These transitions can be attributed to the denaturation of spectrin at ca. 50 °C; bands 2.1 (Ankyrin), 4.1, and 4.2 proteins at ca. 57 °C with a shoulder at 55 °C (hereafter referred to as Band 2–4); the cytoplasmic domain of Band 3 at ca. 63 °C; and Hb endothermic transition at ca. 72 °C, followed by exothermic transition associated with Hb aggregation.

Calorimetric profiles of fresh RBCs from CLL patients, untreated and treated with Obinutuzumab/Venetoclax or Ibrutinib, exhibited the same number of thermal transitions as those of healthy donors (Figure 7). These profiles were subsequently compared by analyzing the transition temperatures (T_m_) and the excess heat capacity (c_P_^ex^) of the successive transitions (Table 3).

RBCs from untreated CLL patients exhibited alterations in the thermal behavior of erythrocyte proteins compared with the control group. The spectrin transition showed slightly elevated T_m_ (+0.6 °C), accompanied by a significant increase in c_P_^ex^ (Figure 7B, Table 3). Transitions associated with Bands 2.1, 4.1, and 4.2 displayed lower T_m_ and elevated c_P_^ex^, reflecting altered membrane–cytoskeleton interactions (Table 3). Band 3 presented stable T_m_ but increased c_P_^ex^, suggesting modified interactions with the surrounding lipid bilayer or adjacent cytoskeletal proteins (Figure 7B, Table 3). Hemoglobin in untreated patients was slightly destabilized, and c_P_^ex^ was reduced (Figure 7A, Table 3). No statistically significant difference was observed in the total enthalpy (ΔH_cal_) relative to the control group.

RBCs from patients treated with the Obinutuzumab/Venetoclax combination displayed further deviations (Figure 7C). The spectrin transition and the shoulder of Band 2–4 transition, typically observed around 55 °C, were not clearly resolved, suggesting either overlap with adjacent transitions or substantial structural perturbation. While the main transition corresponding to Band 2.1, 4.1, 4.2, and Band 3 transition did not show statistically significant shifts in T_m_ relative to controls, they exhibited markedly elevated c_P_^ex^ values (Figure 7D, Table 3). Although the T_m_ of the hemoglobin transition remained similar to controls (71.9 ± 0.08 °C), its c_P_^ex^ was higher (0.83 ± 0.11) (Figure 7C, Table 3). The calorimetric enthalpy (ΔH_cal_) had a significantly higher value than the control, suggesting greater energy requirements for protein unfolding (Table 3). These results indicate that Obinutuzumab/Venetoclax treatment may exacerbate membrane stability.

In contrast, RBCs from patients treated with Ibrutinib displayed thermodynamic properties that closely resembled those of healthy controls (Figure 7E). Spectrin and Band 2–4 transitions were clearly resolved and exhibited only minor deviations in T_m_ and c_P_^ex^, suggesting partial normalization of cytoskeletal dynamics (Figure 7F, Table 3). Hb stability was preserved (T_m_ = 72.1 ± 0.11 °C), and the corresponding c_P_^ex^ remained close to the normal range (0.68 ± 0.05), further supporting the hypothesis that Ibrutinib may exert a protective effect on RBC structural integrity and function.

## 3. Discussion

RBC oxygen transport critically depends on the deformability of the plasma membrane, which, together with the underlying cytoskeleton, allows reversible shape deformations necessary for efficient microcirculatory flow [27,28]. RBCs from healthy donors exhibit significantly greater deformability than RBCs in their diseased state [19]. While CLL primarily affects lymphoid cells, the impact of the disease and its treatments on RBC mechanics remains unclear.

Our study provides a nanomechanical and thermodynamic characterization of RBCs from patients with CLL, highlighting significant alterations associated with the disease and its targeted treatments.

### 3.1. RBC Nanostructural and Nanomechanical Alteration in CLL

The observed decrease in membrane roughness and elasticity in RBCs of untreated CLL patients compared with RBCs from healthy donors likely reflects nanoscale alteration in the lipid bilayer and underlying cytoskeletal framework. These AFM-derived parameters are susceptible to changes in membrane protein composition and cytoskeletal anchoring [29]. Multiple systemic factors associated with CLL may contribute to these biophysical modifications. RBCs circulate in an environment influenced by interactions with malignant lymphocytes, a dysregulated immune system, and systemic oxidative stress inherent to hematologic malignancies [30]. Beyond impaired erythropoiesis and immune-mediated destruction, recent studies suggest that CLL may induce biophysical and rheological changes in RBCs [16,31]. A key mechanism involves the chronic inflammatory tumor microenvironment, where malignant B cells and stromal components release cytokines (e.g., IL-6, TNF-α), chemokines, ROS, and growth factors [32,33]. RBCs, being particularly vulnerable to elevated ROS levels, experience lipid peroxidation and protein oxidation, affecting spectrin, actin, and Band 3 [34]. Oxidative modifications weaken cytoskeletal interactions, compromise membrane stability, and reduce deformability [35], thereby explaining the nanomechanical abnormalities observed in untreated CLL.

### 3.2. Treatment-Related Changes in RBC Properties

Treatment effects on RBC properties reflect the fundamentally different mechanisms of action of the two therapeutic regimens. Obinutuzumab/Venetoclax induces direct cytotoxicity and apoptosis, whereas Ibrutinib modulates signaling pathways and exerts anti-inflammatory effects. Consistent with these divergent mechanisms, we observed distinct patterns of RBC biophysical alterations in our study.

Venetoclax, a BH3 mimetic, blocks the anti-apoptotic BCL-2 protein, thereby removing survival signals in erythroid cells and promoting erythrocyte death by antagonizing Bcl-XL [36]. Healthy mature RBCs express BCL-2, which contributes to their longevity; thus, BCL-2 inhibition can lead to anemia. This aligns with our hematological data (Table 1), which shows reduced RBC counts and hemoglobin levels in the Obinutuzumab/Venetoclax cohort. Anemia, in turn, increases oxidative stress on RBCs, potentially impairing their deformability.

Patients receiving Obinutuzumab/Venetoclax demonstrated normalization of membrane roughness, but an increase in membrane stiffness compared to both untreated patients and healthy controls. This dissociation suggests that although surface topography may improve with therapy, underlying biomechanical abnormalities persist or worsen, consistent with the findings of Longo et al., who reported that membrane roughness and Young’s modulus are not correlated and reflect distinct biophysical features [23].

Obinutuzumab can also induce substantial production of pro-inflammatory cytokines, particularly IL-6 and IL-8 [37,38], which may influence RBC oxidative status and mechanics. Although there is no direct evidence that Obinutuzumab/Venetoclax alters RBC lipids or membrane proteins, oxidative stress-mediated membrane damage has been well documented for other anticancer therapies [39]. In line with these observations, our recent study demonstrated abnormalities in RBC aggregation and morphology in CLL patients treated with Obinutuzumab/Venetoclax, indicating that rheological dysfunction persists despite therapy [16].

In contrast, treatment with the BTK inhibitor Ibrutinib resulted in a partial restoration of both membrane roughness and elasticity to normal levels. This improvement may reflect Ibrutinib’s broader impact on cytoskeletal regulatory pathways and inflammatory processes. BTK inhibition disrupts BCR-associated downstream signaling pathways, including PI3K-AKT, PLCγ2, and Rho GTPases, which regulate B cell actin dynamics and membrane structural integrity [40,41]. Although RBCs lack nuclei, they retain active kinase signaling and post-translational regulatory mechanisms that modulate cytoskeletal and membrane properties [42]. Phosphorylation of cytoskeletal proteins has been shown to influence RBC deformability and mechanical stability [43].

Ibrutinib reduces circulating cytokines such as IL-6, TNF-α, and IL-8 [44], thereby decreasing oxidative and mechanical stress that would otherwise impair RBC integrity. The absence of elevated RDW or NRBCs in our hematological data confirms that the improved RBC biomechanics are not due to regenerative reticulocytosis but instead reflect treatment-associated stabilization of membrane architecture.

These findings highlight the distinct effects of the two CLL therapies: while both treatment regimens improved surface morphology to varying degrees, only Ibrutinib was associated with concomitant restoration of membrane mechanics. This underscores the importance of integrating topographical (Rrms) and mechanical (Ea) measurements to detect treatment-specific alterations in RBC properties.

### 3.3. Adhesive Forces in CLL RBCs

Another important finding of our investigation is the significantly increased adhesive forces in RBCs from untreated CLL patients and those treated with Obinutuzumab/Venetoclax compared to healthy controls and Ibrutinib-treated patients. These altered adhesion properties probably result from CLL-associated metabolic and oxidative imbalances, as well as drug-induced membrane modifications. They reflect changes in membrane surface chemistry and mechanical properties that influence interaction with the AFM cantilever. Such effects may arise from disruptions in membrane lipid composition, cytoskeletal coupling, and surface protein distribution, phenomena commonly described in various hematologic disorders [45,46,47]. In addition, altered interactions between membrane adhesion receptors such as Lu/BCAM and the underlying cytoskeleton have been shown to increase RBC adhesion in spectrin deficiency disorders [48]. The membrane changes may contribute to the increased adhesiveness of RBCs in CLL, potentially promoting stronger cell–cell interactions. These findings align with our previous studies, showing enhanced aggregation behavior in CLL RBCs. Cross-correlation of rheological parameters and AFM-derived adhesion forces revealed a strong positive correlation between the aggregation area index, obtained via microfluidic analysis [16], and the adhesion force. The Pearson correlation coefficient was the highest in Obinutuzumab/Venetoclax-treated patients (r = 0.91), compared with Ibrutinib-treated (r = 0.72) and untreated (r = 0.65) patients, likely reflecting the smaller sample size.

Oxidative damage, further exacerbated by anticancer therapies, represents an important contributor to altered RBC adhesion. ROS arising from leukemic metabolism and drug-induced stress can oxidize membrane lipids and cytoskeletal proteins, leading to lipid peroxidation, crosslinking of spectrin, and clustering of membrane proteins such as Band 3 [49]. Band 3 is a key structural component linking the RBC membrane to spectrin via protein 4.2 and ankyrin [50]. These oxidative modifications compromise membrane resilience, expose adhesion-promoting sites and phosphatidylserine residues, and enhance cell–cell and cell–matrix interactions [51,52,53].

Increased RBC adhesion can impair microvascular flow and promote endothelial activation [54], contributing to anemia, tissue hypoxia, and vascular complications frequently seen in CLL patients [55]. Under Obinutuzumab/Venetoclax therapy, enhanced oxidative stress associated with anemia and membrane fragility could exacerbate hemolytic processes or reduce RBC lifespan. RBCs from Ibrutinib-treated patients exhibited intermediate adhesion values approaching those of healthy controls. This likely reflects partial restoration of membrane homeostasis, probably through modulation of cytoskeletal and adhesion-related signaling pathways (as shown in leukocyte studies [56,57]) and reduced oxidative stress, though direct evidence in RBCs remains to be established.

Together, these findings suggest that the adhesive and mechanical properties of CLL RBCs reflect a broader dysregulation of membrane homeostasis involving oxidative stress, lipid remodeling, and cytoskeletal detachment. The increased adhesion strength observed by AFM thus likely represents a composite biophysical marker of oxidative membrane injury and disturbed lipid–cytoskeletal coupling, with potential implications for microcirculatory flow.

### 3.4. Thermodynamic Behavior of RBCs in CLL

DSC provides complementary insights into the structural stability of key erythrocyte proteins and their intermolecular interactions in CLL.

RBCs from untreated CLL patients exhibited a moderate upshift in the transition temperature of spectrin, accompanied by an increased excess heat capacity, reflecting altered spectrin unfolding and interaction with other components of the cell membrane. The lower T_m_ and elevated c_P_^ex^ values for Band 2–4 (ankyrin, proteins 4.1 and 4.2) indicate perturbations in the cytoskeleton–membrane linkage, consistent with decreased roughness and increased stiffness observed by AFM. Hemoglobin displayed a mild decrease in the Tm and c_P_^ex^, suggesting partial destabilization of its quaternary structure, possibly due to oxidative modifications.

The Obinutuzumab/Venetoclax–treated group demonstrated even greater deviations. The spectrin transition peak and the shoulder of the Band 2–4 transition were poorly resolved, potentially due to neighboring transitions or decreased unfolding cooperativity. Despite normalization of membrane roughness, the elevated c_P_^ex^ values for Band 2–4 and Band 3 transitions, together with increased calorimetric enthalpy, indicate altered protein–protein and protein–lipid interactions, possibly related to oxidative stress–driven membrane remodeling and changes in cytoskeletal anchoring.

In contrast, the thermodynamic behavior of RBCs from Ibrutinib-treated patients closely resembled that of healthy controls. Spectrin and Band 2–4 transitions were well resolved, with minor deviations in T_m_ and c_P_^ex^, whereas hemoglobin unfolding behavior remained unchanged. These findings are consistent with partial restoration of RBC nanomechanics following Ibrutinib treatment.

DSC data indicate that CLL and its therapies induce distinct patterns of RBC thermodynamic behavior. The heat capacities and enthalpy change in major RBC proteins reflect perturbations in their binding interactions, cytoskeletal anchoring, and post-translational modifications. For example, modifications of the cytoplasmic domain of Band 3, caused by oxidative stress or phosphorylation, may compromise its interaction with the spectrin–actin skeleton, thereby affecting membrane stability and protein interactions [58,59].

Alterations in protein unfolding transitions correlate with nanomechanical properties of RBCs, indicating that the energy landscape of cytoskeletal and membrane protein interactions shapes cell deformability and adhesive behavior. Thus, DSC reinforces AFM, providing a thermodynamic characterization of RBC membrane remodeling and the impact of targeted therapies in CLL.

## 4. Materials and Methods

### 4.1. Selection of Patients and Ethics Statement

This research study was approved by the Ethics Committee of the Institute of Biophysics and Biomedical Engineering, Bulgarian Academy of Sciences (Approval No. 378HД, 26 March 2024), and conducted in accordance with the principles of the Declaration of Helsinki (1975), revised in 2013, for research involving human subjects. Informed consent was obtained from all participants before enrollment.

A total of twenty-seven patients diagnosed with CLL were enrolled in the study. Diagnosis was based on established clinical criteria in accordance with current guidelines [60,61]. All patients were admitted to the National Specialized Hospital for Active Treatment of Hematological Diseases, Sofia, Bulgaria. 

The study population included: eight treatment-naïve patients (mean age: 60.5 ± 12.3 years; range: 37–73 years), eight patients receiving Ibrutinib (mean age: 64.13 ± 9.2 years; range: 51–77 years), and eleven patients receiving Obinutuzumab/Venetoclax combination therapy (mean age: 69.6 ± 7.5 years; range: 53–80 years).

The study also included a control group of 17 healthy individuals (11 females and 6 males; mean age: 56.6 ± 8.8 years; range: 37–69 years). None of the volunteers were smokers, had received any medical treatment, and none had a history of CLL, other oncohematological disorders, hereditary predisposition, or other chronic diseases. 

### 4.2. Blood Collection and Sample Preparation

Peripheral blood was collected from CLL patients during routine clinical visits and from healthy volunteers following an overnight fast. Blood samples were drawn into two 6 mL K_2_EDTA Vacutainer tubes (Becton Dickinson and Company, Franklin Lakes, NJ, USA) from each participant. RBCs were isolated by centrifugation at 800× *g* for 15 min at 4 °C using a Sigma 2-16KL centrifuge (Merck, Osterode, Germany), and the plasma and buffy coat (which contains white blood cells) were carefully removed. The remaining RBC fraction was resuspended and washed three times in phosphate-buffered saline (PBS), consisting of 10 mM sodium phosphate (pH 7.2), 140 mM NaCl, and 1 mM EDTA prior to AFM and DSC analyses.

For AFM analysis, washed RBCs were diluted in PBS to a final hematocrit of 4%. Sample smears were prepared in duplicate following the protocol described by Dinarelli et al. [22]. Briefly, RBC suspensions (15 μL) were diluted 1:1 (*v*/*v*) with autologous plasma to minimize osmotic stress. According to Longo et al. (2025), plasma preserves cell morphology and ensures a homogeneous erythrocyte distribution on a glass slide, facilitating AFM measurements [23]. The diluted RBC suspensions were carefully applied onto poly-L-lysine-coated coverslips to promote cell adhesion and prevent cell displacement during drying and AFM imaging. No chemical fixation (e.g., glutaraldehyde) was used. Following 30 min of air-drying under controlled humidity, samples were subjected to optical and AFM imaging.

### 4.3. Optical Microscopy

The morphological types of RBCs from control groups and CLL patients were assessed using a 3D optical profiler (Zeta-20, Zeta Instruments, Milpitas, CA, USA). Images were acquired with a 50× objective lens. Typically, approximately 250–270 cells were analyzed per sample. All experiments were performed at room temperature.

### 4.4. Atomic Force Microscopy

AFM (MFP-3D, Asylum Research, Oxford Instruments, Santa Barbara, CA, USA) was employed to obtain high-resolution topographical images and to assess the morphometric and nanomechanical properties of RBCs. Measurements were performed in contact mode at 22–24 °C to reduce thermal drift and maintain stable probe–sample interactions. Silicon AFM probes (Nanosensors, type qp-Bio) with a 50 kHz resonance frequency 0.3 N/m nominal spring, and a conical tip with a nominal radius of 8 nm were used. Morphometric and nanoindentation measurements were carried out on 20–25 cells to ensure reproducibility and statistical robustness.

#### 4.4.1. Morphometric Analysis

Imaging of RBC surface topology was performed in air using contact mode. Morphometric characterization, specifically membrane roughness, was carried out using Igor Pro 6.37 software. Roughness analysis was conducted on multiple square areas of fixed size (2.0 × 2.0 µm^2^) for each RBC. To minimize artifacts arising from overall cell curvature or distortion, a first-order flattening procedure was applied to all selected areas. For biconcave discocytes, measurements were performed in the peripheral regions of the cell. For spiculocytes and spherocytes, however, central membrane regions were analyzed due to their altered geometry. Membrane surface roughness was quantified as root mean square roughness (R_rms_) of the height distribution, calculated as follows:$$R_{\mathrm{rms}}\;=\;\sqrt{\sum_{ì=1}^{N}\frac{{\left(z_{i}-z_{m}\right)}^{2}}{\left(N-1\right)}}$$
where *N* is the total number of points, *z_i_* is the height of the *i*th point, *z_m_* is the mean height, and *N* is the total number of points.

#### 4.4.2. Young’s Modulus Measurements

After topographical imaging, AFM force–distance curves were recorded on the same cells. A sterile PBS (40 µL) was applied to the RBC sample on a glass coverslip to ensure proper probe–sample interaction, with an additional 5 µL droplet placed on the AFM tip to prevent air pocket formation. Force-distance curves were acquired in contact mode at room temperature. Young’s modulus (Ea) was extracted by fitting the curves to the Hertz-Sneddon model using IgorPro 6.37 software. Force mapping was performed on a 16 × 16 grid at a scanning speed of ~3.7 µm/s. The AFM tip was calibrated before each measurement session on a clean glass substrate using the built-in Igor Pro calibration routine. The maximum force that did not induce a nonlinear response ranged from 10 to 20 nN, depending on the tip calibration and the measured sample.

Young’s modulus (Ea) was determined using the Hertz–Sneddon model for a conical indenter:$$F(\delta )=\frac{2Etan\left(\alpha \right)}{\pi \left(1-\nu^{2}\right)}\delta^{2}$$
where *F*(*δ*) is the applied force as a function of indentation depth *δ*, *E* is the apparent Young’s modulus, *α* is the half-opening angle of the conical tip, *ν* is the Poisson ratio (assumed to be 0.5 for biological samples), and *δ* is the indentation depth.

#### 4.4.3. Adhesion Force Measurements

Adhesion (pull-off) forces were collected simultaneously with indentation data from the retract segments of the force–distance curves. These values capture non-specific interactions between the AFM tip and the RBC membrane, including contributions from membrane proteins, glycocalyx components, and surface charge. Adhesion maps were analyzed using Igor Pro, and mean adhesion forces were determined for each cell from all force curves within a map.

### 4.5. DSC Measurements

DSC measurements were performed using a DASM-4 scanning calorimeter (Biopribor, Pushchino, Russia). Scans were conducted at a rate of 1.0 °C/min over a temperature range of 30–90 °C. A 2 atm overpressure was applied over the liquids in the calorimetric cells throughout the scans to prevent degassing during heating.

Before each measurement, RBCs were washed three times with PBS and adjusted to a Hb concentration of 8 mg/mL. The reversibility of the thermal transitions was assessed by recording a second calorimetric scan immediately after cooling from the first. As the transitions were irreversible, the reheating thermogram served as the instrumental baseline. Heat capacity differences between the initial and final states were corrected using a linear chemical baseline, as described previously [62]. The resulting thermograms were normalized to the Hb concentration, smoothed, and analyzed using the OriginPro 2018 software package. The thermodynamic parameters, including the denaturation temperature (T_m_) and the excess heat capacity (c_P_^ex^) corresponding to the resolved thermal transitions, and the total calorimetric enthalpy (ΔH_cal_), were determined from the resulting scans.

### 4.6. Clinical and Hematological Indices

Clinical and hematological indices were obtained from the National Specialized Hospital for Active Treatment of Hematological Diseases, Sofia, Bulgaria. Biochemical analyses were performed using the Architect c4000 and Beckman Coulter AU480 analyzers (Chaska, MN, USA), and hematological analyses with the Siemens ADVIA 2120i (Tarrytown, NY, USA) and Dirui BF-7200 Plus systems (Dirui Industrial Co., Ltd., Changchun, China). All procedures followed standard clinical protocols.

### 4.7. Statistical Analysis

Data are presented as Me (Q1; Q3), where Q1 and Q3 represent the first and third quartiles, respectively. The Shapiro–Wilk test was used to assess data normality. As several variables violated the assumptions of normality and homogeneity of variance, non-parametric tests were applied. Differences among the four independent groups were evaluated using the Kruskal–Wallis test, followed by Dunn’s post hoc test. Differences were considered statistically significant at *p* < 0.05.

## 5. Conclusions

This study provides novel biophysical insights into the membrane roughness, nanomechanical properties, and thermodynamic alterations of RBCs from CLL patients. It also highlights the distinct effects of targeted therapies on erythrocyte integrity. RBCs from untreated CLL patients exhibit reduced membrane roughness, increased stiffness and adhesion force, and altered thermal unfolding profiles of major cytoskeletal and membrane proteins, indicating compromised deformability and disrupted membrane–cytoskeleton coupling. While Obinutuzumab/Venetoclax treatment restores membrane surface roughness, it does not fully correct membrane stiffness, its adhesion force or the thermal unfolding profiles of the major cytoskeletal and membrane proteins, suggesting persistent biomechanical dysfunction. In contrast, Ibrutinib therapy normalizes the roughness, nanomechanical properties, and calorimetric profiles, indicating a more comprehensive restoration of RBC properties. Overall, the integrated AFM and DSC analyses highlight the systemic impact of CLL on non-malignant blood cells and the potential of nanomechanical and thermodynamic RBC parameters as complementary indicators for disease monitoring and therapeutic response.

## Figures and Tables

**Figure 1 ijms-27-00353-f001:**
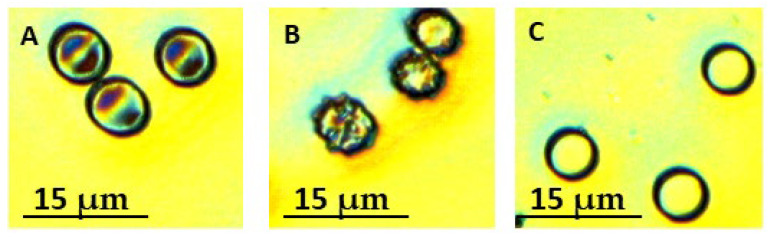
Representative 2D optical microscopy images illustrating the main RBC morphological phenotypes observed in the study: (**A**) biconcave discocytes with the characteristic central pallor; (**B**) spiculated cells exhibiting multiple membrane protrusions; and (**C**) spherocytes displaying a spherical shape and loss of central pallor. The images were captured using a 50× magnification objective lens.

**Figure 2 ijms-27-00353-f002:**
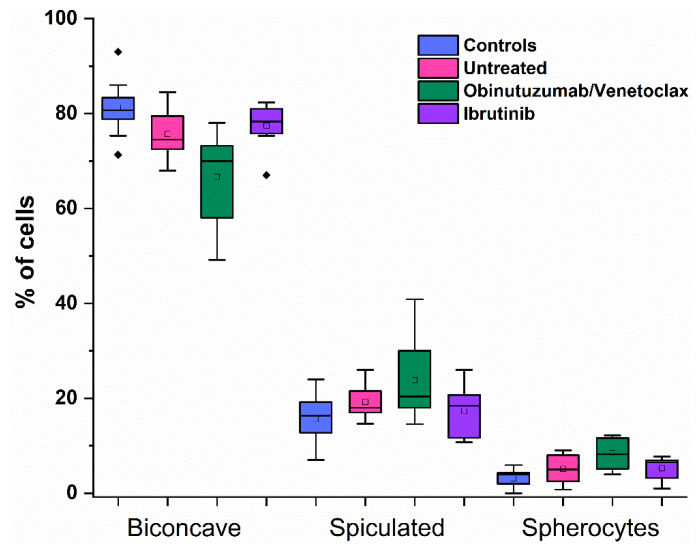
Box plot of the relative proportions of RBC morphological classes—biconcave (discocytes), spiculated, and spherocytes, determined by optical microscopy. Data are presented for healthy controls (*n* = 17), untreated CLL patients (*n* = 8), and CLL patients treated with Obinutuzumab/Venetoclax (*n* = 11) or Ibrutinib (*n* = 8). For each sample, RBC morphology was evaluated by counting a total of 250–270 cells, and the proportion of each morphological class was normalized to the total number of cells counted per donor. Box plots display the median and interquartile range, with whiskers representing variability across groups.

**Figure 3 ijms-27-00353-f003:**
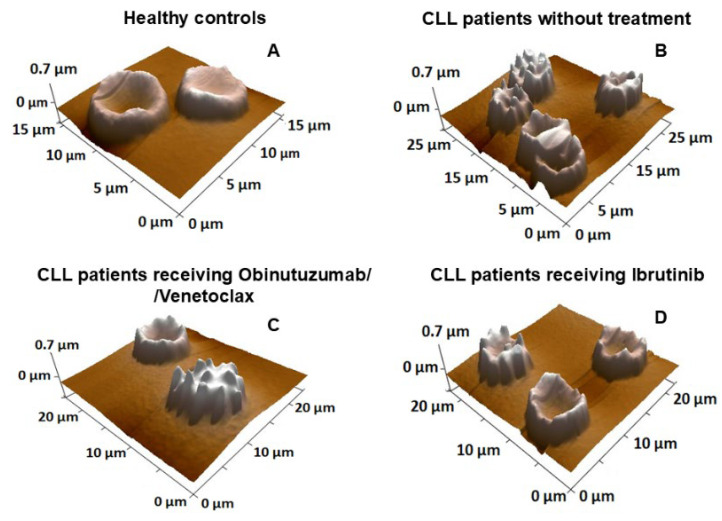
Representative 3D AFM images of RBCs isolated from healthy controls (**A**), untreated CLL patients (**B**), CLL patients treated with the Obinutuzumab/Venetoclax combination (**C**), and CLL patients treated with Ibrutinib (**D**), illustrating the corresponding morphological types in the studied groups.

**Figure 4 ijms-27-00353-f004:**
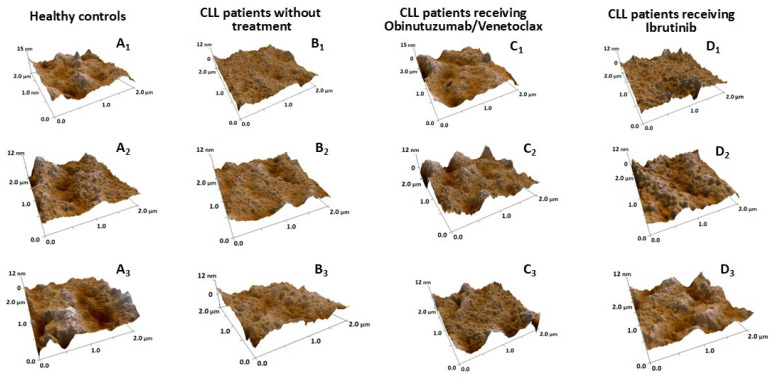
Representative high-resolution 3D AFM images of RBCs. Panels show samples from three independent donors/patients per group, illustrating intra-group variability in membrane architecture: (**A_1_**–**A_3_**) healthy donors; (**B_1_**–**B_3_**) untreated CLL patients; (**C_1_**–**C_3_**) CLL patients treated with Obinutuzumab/Venetoclax; and (**D_1_**–**D_3_**) CLL patients treated with Ibrutinib. Images were acquired in contact mode with a scan size of 2 × 2 μm.

**Figure 5 ijms-27-00353-f005:**
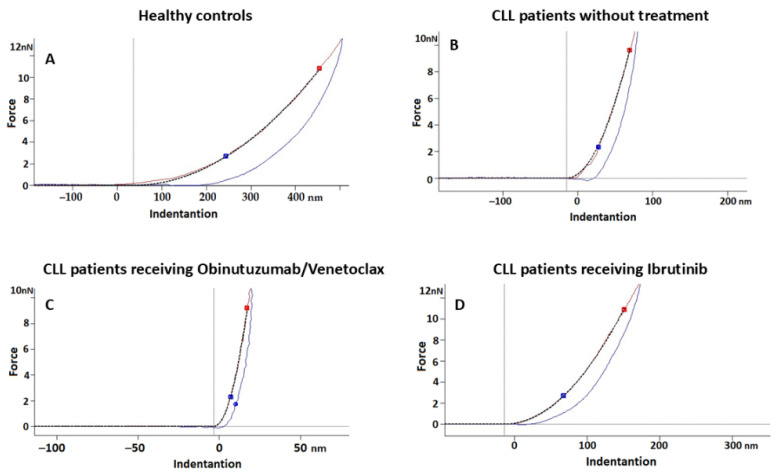
Representative force–indentation curves obtained from RBC samples of healthy controls (**A**), untreated CLL patients (**B**), CLL patients treated with the Obinutuzumab/Venetoclax combination (**C**), and CLL patients treated with Ibrutinib (**D**). The different lines and symbols represent individual RBC measurements, automatically generated by the plotting software. Each force–indentation curve, obtained from AFM data using Igor Pro, plots cantilever deflection (force) versus tip–sample distance, with the slope reflecting cell stiffness.

**Figure 6 ijms-27-00353-f006:**
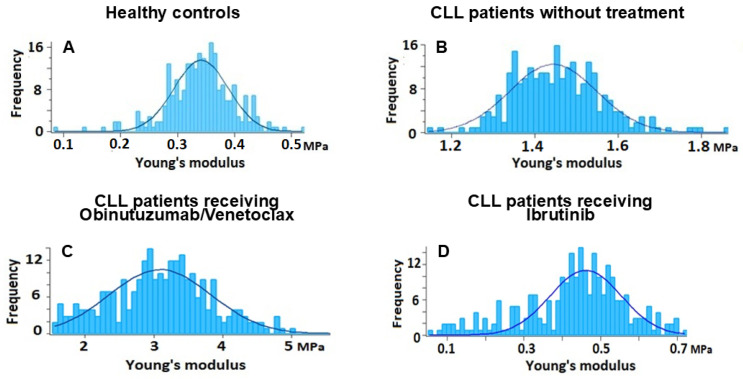
Histograms of Young’s modulus distribution of the membrane of RBCs isolated from healthy patients (**A**), untreated CLL patients (**B**), CLL patients receiving Obinutuzumab/Venetoclax (**C**), and CLL patients receiving Ibrutinib (**D**).

**Figure 7 ijms-27-00353-f007:**
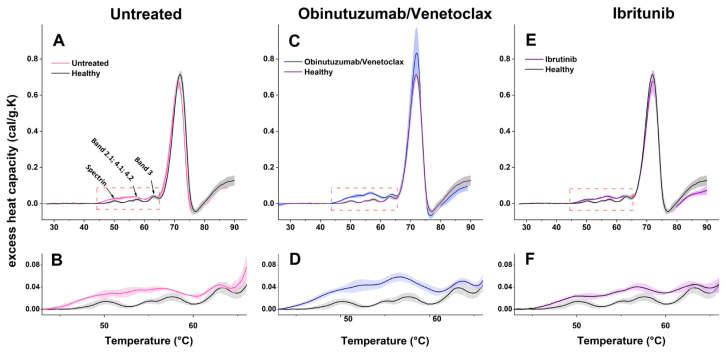
DSC profiles (mean value ± SD) of RBCs from untreated patients (**panel A**, red line), Obinutuzumab/Venetoclax-treated patients (**panel C**, blue line), and Ibrutinib-treated patients (**panel E**, violet line), plotted together with the average thermograms from healthy controls (black line). Panels (**B**,**D**,**F**) provide enlarged views of the low-temperature range (~45–65 °C, enclosed by a dashed red box) of the respective thermograms in panels (**A**,**C**,**E**), highlighting endothermic transitions associated with the denaturation of cytoskeletal and membrane proteins, including spectrin, Band 2–4, and Band 3.

**Table 1 ijms-27-00353-t001:** Clinical (age, gender, and Rai stage), and laboratory indices (RBC count; hemoglobin, Hb; hematocrit, Ht; mean corpuscular volume, MCV; mean corpuscular hemoglobin, MCH; mean corpuscular hemoglobin concentration, MCHC; red blood cell distribution width, RDW; total bilirubin; white blood cell count, WBC; lymphocytes count; platelet count; and NRBCs/erythroblasts) determined for healthy controls and CLL patients. Data are presented as Me (Q1; Q3).

Parameters	Reference Value	Studied Groups				
		Healthy Controls (*n* = 17)	Untreated CLL Patients (*n* = 8)	CLL Patients Receiving Obinutuzumab/ Venetoclax (*n* = 11)	CLL Patients Receiving Ibrutinib (*n* = 8)	*p*-Value (Kruskal–Wallis Test)
Age (years)	-	58 (49.0; 65.5) ^a^	65 (49; 70.8) ^a^	72 (64.5; 74.8) ^b^	62.5 (54.3; 73.8) ^ab^	0.04
Gender (F/M)		11/6	3/5	5/6	3/5	
Rai stage			0	1–4	1–4	
Onset of disease (years)			2.5 (0.75; 9.0)	4.0 (3.0; 7.0)	8.0 (5.0; 13.0)	
Treatment duration (years)				2.0 (1.0; 2.0)	4.0 (4.0; 5.0)	
RBC count (T/L)	4.60–6.20	4.45 (4.38; 4.79) ^a^	5.16 (4.68; 5.26) ^b^	4.62 (4.46; 4.82) ^a^	4.91 (4.67; 5.09) ^a^	0.048
Hb (g/L)	140.00–180.00	143.5 (140; 151) ^a^	148.5 (141; 160) ^a^	140.0 (120; 148) ^b^	141 (138; 147) ^a^	0.009
Ht (L/L)	0.40–0.54	0.44 (0.42; 0.47)	0.44 (0.42; 0.46)	0.41 (0.37; 0.43)	0.43 (0.41; 0.46)	0.06
MCV (fl)	80.00–95.00	89.8 (87.4; 93.0)	87.9 (85.8; 93.8)	88.1 (82.9; 93.6)	89.2 (85.9; 91.0)	0.88
MCH (pg/L)	27.00–32.00	30.1 (29.1; 31.4)	30.3 (28.5; 31.5)	301 (27.7; 31.3)	29.7 (27.5; 30.2)	0.41
MCHC (g/L)	320.00–360.00	343.0 (327; 349)	339.5 (324; 345)	337.0 (328; 341)	333 (306; 344)	0.43
RDW%	11.60–14.80	13.2 (12.7; 14.3)	13.5 (13.2; 15.4)	13.9 (13.3; 14.7)	14.1 (13.5; 14.6)	0.95
WBC	3.50–10.50	5.8 (5.4; 6.6)	21.2 (10.1; 71.2) *	3.89 (2.82; 4.20)	6.1 (4.8; 8.1)	0.015
Total bilirubin (μmol/L)	3.40–20.50	16.3 (7.4; 18.5)	12.0 (8.8; 37.9)	13.0 (9.5; 21.5)	13.4 (11.3; 18.4)	0.73
Lymphocytes (ABS)	1.10–3.80	1.84 (1.79; 2.14)	15.8 (5.4; 62.4) *	1.2 (1.0; 1.5)	1.8 (1.1; 3.4)	<0.0001
Platelet count ×10^9^/L		289 (192; 376)	200 (168; 239)	161 (131; 214)	154 (142; 195)	0.43
NRBCs/erythroblasts (g/L)	0.00–0.20	Not measured	None established	None established	None established	-

Statistically significant differences between groups are indicated by superscripts (Dunn’s post hoc test, *p* < 0.05). Groups that share the same letter are not significantly different. * Extreme value observed in an untreated CLL.

**Table 2 ijms-27-00353-t002:** Membrane Roughness (Rrms), Elastic Modulus (Ea), and Adhesive Forces of RBCs from the Studied Groups. For each individual, 20–25 cells were analyzed. Data are presented as Me (Q1; Q3).

Sample (*n*)	Rrms (nm)	Ea (MPa)	Adhesive Forces (pN)
RBCs from healthy donors (17)	4.81 (4.07; 5.29) ^a^	0.336 (0.28; 0.43) ^a^	266.5 (177; 321.8) ^a^
RBCs from untreated Patients (8)	3.55 (2.53; 4.03) ^b^	1.3 (1.02; 2.03) ^b^	351.13 (283.9; 387.3) ^b^
RBCs from Obinutuzumab/Venetoclax-treated CLL patients (11)	4.28 (3.19; 6.40) ^a^	3.53 (2.76; 3.75) ^c^	350 (248; 513.3) ^b^
RBCs from Ibrutinib-treated CLL patients (8)	4.19 (3.02; 4.71) ^a^	0.493 (0.31; 0.87) ^a^	261.14 (211.6; 383.2) ^a^
*p*-value (Kruskal–Wallis test)	0.007	<0.001	0.002

Statistically significant differences between groups are indicated by superscripts (Dunn’s post hoc test, *p* < 0.05). Groups that share the same letter are not significantly different.

**Table 3 ijms-27-00353-t003:** Transition temperature (Tm) and excess heat capacities (c_P_^ex^) of successive endothermic transitions in RBCs from healthy donors, untreated CLL patients, and patients receiving Obinutuzumab/Venetoclax combination or Ibrutinib. Data are presented as Me (Q1; Q3). nd—not detected.

Parameters	RBCs from Healthy Controls	RBCs from Untreated Patients	RBCs from Patients Treated with Obinutuzumab/ Venetoclax	RBCs from Patients Treated with Ibrutinib	*p*-Value (Kruskal–Wallis Test)
T_m_^spectrin^ (°C)	50.1 (50.07; 50.2)	50.7 (50.63; 50.81)	nd	49.88 (49.7; 49.92)	0.52
c_P_^spectrin^ (cal/gK)	0.014 (0.013; 0.016) ^a^	0.029 (0.026; 0.030) ^b^	nd	0.024 (0.023; 0.025) ^b^	<0.001
T_m_^Band2–4^ (°C)	55.2 (55.1; 55.43) ^a^/ 57.5 (57.35; 57.68) ^a^	54.05 (53.98; 54.22) ^b^/ 56.48 (56.33; 56.76) ^b^	nd/ 56.73 (56.63; 56.89) ^a^	54.5 (53.61; 54.83) ^a^/ 56.82 (56.41; 57.36) ^a^	<0.001
c_P_^Band2–4^ (cal/gK)	0.0148 (0.0137; 0.0158) ^a^/ 0.023 (0.020; 0.024) ^a^	0.034 (0.031; 0.035) ^b^/ 0.037 (0.035; 0.040) ^b^	nd/ 0.059 (0.056; 0.062) ^c^	0.032 (0.030; 0.034) ^b^/ 0.041 (0.039; 0.044) ^b^	<0.001
T_m_^Band3^ (°C)	63.29 (63.16; 63.46)	63.05 (62.98; 63.16)	63.7 (63.68; 63.87)	63.11 (62.78; 63.26)	0.76
c_P_^Band3^ (cal/gK)	0.038 (0.035; 0.041) ^a^	0.043 (0.040; 0.046) ^b^	0.050 (0.047; 0.053) ^c^	0.045 (0.042; 0.047) ^b^	0.03
T_m_^Hb^ (°C)	71.91 (71.77; 72.01)	71.27 (71.19; 71.38)	71.92 (71.02; 71.28)	72.13 (71.99; 72.27)	0.85
c_P_^Hb^ (cal/gK)	0.718 (0.707; 0.73) ^a^	0.677 (0.668; 0.689) ^b^	0.839 (0.819; 0.859) ^c^	0.688 (0.671; 0.7) ^b^	<0.001
ΔH_cal_ (cal/g)	3.84 (3.77; 3.97) ^a^	3.94 (3.82; 4.05) ^a^	4.71 (4.57; 4.85) ^b^	3.99 (3.87; 4.11) ^a^	0.01

Statistically significant differences between groups are indicated by superscript letters (Dunn’s post hoc test, *p* < 0.05); groups sharing the same letter are not significantly different.

## Data Availability

The original contributions presented in this study are included in the article. Further inquiries can be directed to the corresponding author.

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
