# Peer review of "Nanomechanical and Thermodynamic Alterations of Red Blood Cells in Chronic Lymphocytic Leukemia: Implications for Disease and Treatment Monitoring"

_ijms, 2025, doi:10.3390/ijms27010353_

Round 1

Reviewer 1 Report

Comments and Suggestions for Authors

This paper presents a study to investigate the nanomechanical properties of red blood cells in chronic lymphocytic leukemia patients by AFM as well as the cellular thermodynamic properties. The topic is important, as mechanical cues are crucial for cellular activities and diseases. However, after reading the manuscript, I have several concerns about the methodology of AFM in the paper, which are following:

  1. I noticed that cells air-dried and then measured by AFM. Air-drying can significantly cause destructive changes to the cell structures and thus cell mechanics. Therefore, results obtained on air-dried cells can not reflect the real situations of living cells. In fact, the Young’s modulus of cells was much higher (about 1000 times larger) than that measured by other researchers (Biophys. J., 2006, 90: 2994-3003), and this is because the authors measured air-dried cells in this work and air-dried cells are significantly harder than living cells.
  2. The details of AFM experiments are lacking. For example, the AFM probe tip, probe cantilever spring constant, force curve parameters, how many cells were tested in each experiment, and so on. Besides, typical force curves obtained on different types of cells need to be provided.
  3. AFM is particularly suitable for imaging living cells and measuring their mechanics in the physiological solution. To my taste, the authors need to perform AFM experiments directly on living red blood cells but not on dried cells, and results obtained on living cells are much more faithful. I know that red blood cells are suspended cells, and it is challenging to image them by AFM and you need to use adequate immobilization methods to trap them before AFM imaging, but at least measuring cell mechanics can be performed on living cells.

Overall, I think the AFM experiments of the paper have technical flaws, making the results unreliable. Hence, I do not think the paper has the academic quality to publish in this journal.

Author Response

This paper presents a study to investigate the nanomechanical properties of red blood cells in chronic lymphocytic leukemia patients by AFM as well as the cellular thermodynamic properties. The topic is important, as mechanical cues are crucial for cellular activities and diseases. However, after reading the manuscript, I have several concerns about the methodology of AFM in the paper, which are following:

Comment 1: I noticed that cells air-dried and then measured by AFM. Air-drying can significantly cause destructive changes to the cell structures and thus cell mechanics. Therefore, results obtained on air-dried cells can not reflect the real situations of living cells. In fact, the Young’s modulus of cells was much higher (about 1000 times larger) than that measured by other researchers (Biophys. J., 2006, 90: 2994-3003), and this is because the authors measured air-dried cells in this work and air-dried cells are significantly harder than living cells.

Response 1: We appreciate the reviewer’s concern regarding the possible influence of air-drying on RBC nanomechanics. In AFM studies of red blood cells, there is no standardized sample preparation protocol, and various approaches have been reported, including imaging and mechanical probing of chemically fixed cells (e.g., with glutaraldehyde), as well as native state - live cells (Zachée et al, 1996, Br J Haematol; Abay et al., 2019, Front Physiol; Pi & Cai 2019, Methods Mol Biol;).

It should be emphasized that, in the present study, the RBCs were not glutaraldehyde-treated or chemically fixed (described in section 4.2. Blood Collection and Sample Preparation). Visualization of native RBCs by AFM is limited by the very soft and labile nature of the cell surface (Sergunova et al., 2022, Sensors). Glutaraldehyde fixation is known to induce artifacts, resulting in changes to the membrane surface structure and elasticity (Abay, 2019, Front Physiol). To avoid such artefacts, we used a protocol designed to preserve near-native mechanical properties without chemical modification.

Measuring unfixed RBCs in liquid is technically challenging due to their soft, deformable membrane. Inadequate adhesion (weakly attached cells) frequently results in lateral displacement or detachment during nanoindentation. Therefore, a method was developed in which, to visualize native red blood cells in liquid, it is necessary to prepare an adhesive substrate, for example, poly-L-lysine-coated glass (Sergunova et al., 2022, Sensors). To ensure stable attachment and prevent artefactual movement, samples were briefly air-dried (~30 min). Short-term air-drying has been reported not to cause major destructive changes in RBC membrane morphology (Wu 2009, Micron). This sample preparation strategy has been successfully applied in AFM studies of RBC nanostructure and mechanics by multiple groups (Sergunova 2022, Sensors; Dinarelli 2018, BMC Bioinformatics; Dinarelli 2018, Sci Rep), as well as in our previous work (Strijkova-Kenderova 2022, IJMS; Langari 2022, IJMS; etc.).

When comparing AFM-derived Young’s modulus values across studies, it is essential to match methodological parameters such as AFM model, cantilever stiffness, tip geometry, force model (e.g., Hertz or Hertz-Sneddon), indentation regime, the same type of cells under study, as well as the sample preparation protocol. In the study cited by the reviewer (Rosenbluth 2006, Biophys J, 2006, 90: 2994–3003), different cells were examined (HL-60, Jurkat and neutrophils), using a different AFM system (BioScope AFM, Veeco) and cantilevers (V-shaped, gold-coated silicon nitride, k = 9–11 pN/nm), the Hertz model for Young’s modulus calculation, and a different sample preparation protocol (measurements performed in liquid without adhesion to glass). Therefore, their reported elastic moduli cannot be directly compared to our results obtained under different experimental conditions on native RBCs.

In brief, Young's modulus values obtained for biological samples depend heavily on multiple experimental variables, including tip, sample properties, tip–sample interaction forces, instrument settings, and the model applied to analyse the force curves. Consequently, studies in the literature often report different nanomechanical values for RBCs, even when examining similar cell types. Therefore, ensuring methodological consistency, particularly in probe characteristics and indentation procedures throughout the experiments, is essential. Such consistency enables reliable detection of morphological and mechanical differences that are physically meaningful between the experimental groups.

In our study, unfixed RBCs (Not treated with chemicals like glutaraldehyde or formaldehyde), prepared by adhesion to polylysine-coated glass, were analyzed with an MFP-3D AFM (Asylum Research) using silicon conical probes (k = 0.3 N/m; tip radius 8 nm), and Young’s modulus was calculated using the Hertz-Sneddon model (subsection 4.3. Atomic force microscopy in section 4. Materials and Methods). This protocol follows established procedures optimized for quantitative characterization of RBC nanomechanics, including the approach described by Longo (2025, Methods Protoc.).

Overall, our methodology avoids artefacts associated with chemical fixation, while enabling stable and reproducible AFM force measurements on native RBCs.

Comment 2: The details of AFM experiments are lacking. For example, the AFM probe tip, probe cantilever spring constant, force curve parameters, how many cells were tested in each experiment, and so on. Besides, typical force curves obtained on different types of cells need to be provided.

Response 2: We thank the reviewer for this suggestion. In the revised manuscript (for details, see Section 4.3, Atomic Force Microscopy), we have added full details of the AFM experiments. Measurements were performed using an MFP-3D AFM with silicon qp-Bio probes (50 kHz, 0.3 N/m, conical tip, 8 nm radius). For each sample (donor), 20–25 RBCs were analyzed. Force–distance curves were collected in contact mode in liquid (PBS), at room temperature, and Young’s modulus was calculated using the Hertz–Sneddon model. Adhesion (pull-off) forces were collected simultaneously with indentation data from the retract segments of the force–distance curves. Membrane roughness was also measured from the same cells on several square areas of fixed size (2.0 x 2.0 µm2) for each cell. To address the reviewer’s request, representative force–indentation curves for RBCs from healthy controls, untreated CLL patients, and CLL patients treated with Obinutuzumab/Venetoclax or Ibrutinib have been added to the Supplementary Material (Figure S5) to illustrate the characteristic mechanical responses across groups.

Comment 3: AFM is particularly suitable for imaging living cells and measuring their mechanics in the physiological solution. To my taste, the authors need to perform AFM experiments directly on living red blood cells but not on dried cells, and results obtained on living cells are much more faithful. I know that red blood cells are suspended cells, and it is challenging to image them by AFM and you need to use adequate immobilization methods to trap them before AFM imaging, but at least measuring cell mechanics can be performed on living cells.

 Response 3: We appreciate the reviewer’s suggestion and fully agree that AFM measurements on not dried RBCs in physiological solution are, in principle, desirable. However, as noted by the reviewer, RBCs are non-adherent cells with soft, highly deformable surfaces, which creates major technical challenges for stable immobilization during force spectroscopy.

To minimize osmotic stress and preserve cell morphology, freshly isolated erythrocyte suspensions (15 μL) were diluted 1:1 (v/v) with autologous plasma (for details, please see 4.2. Blood Collection and Sample Preparation). As reported by Longo et al. (2025, Methods Protoc.), the use of plasma ensures optimal preservation of cell morphology. It promotes a homogeneous dispersion of erythrocytes on the glass slide during smear preparation, which is advantageous for AFM measurements.

Quantitative force measurements on freely suspended RBCs in liquid are generally not feasible without robust immobilization, as cells are easily displaced or detached by the AFM tip, resulting in unstable force curves and unreliable estimation of the Young’s modulus (Sergunova et al., 2022; Wu et al., 2009). Therefore, in the present study, we employed a standardized and widely accepted protocol in which samples were briefly air-dried (~30 min) on poly-L-lysine–coated glass to ensure stable adhesion (Longo 2025, Methods Protoc.; Dinarelli 2018, BMC Bioinformatics; Dinarelli 2018, Sci Rep). Importantly, cells were not chemically fixed, avoiding artefacts introduced by glutaraldehyde or other cross-linkers. Following topographic imaging, force–distance curves were acquired in PBS to ensure proper probe–sample interaction.

Nevertheless, we agree with the reviewer that future work should explore approaches enabling measurements on fully living, freely suspended RBCs in physiological solution. While such measurements would require specialized AFM setups, such as microfluidic trapping systems or high-speed AFM in liquid, these devices are not currently available in our laboratory.

Overall, I think the AFM experiments of the paper have technical flaws, making the results unreliable. Hence, I do not think the paper has the academic quality to publish in this journal.

Response: We believe that the above clarifications strengthen the quality and reliability of our study.

Reviewer 2 Report

Comments and Suggestions for Authors

1 The introduction of the article demonstrates a deep understanding of the topic, contains extensive information on the biology of CLL, prognostic markers, and targeted therapy, but it occupies a disproportionate amount of space. The main topic of the research - the nanomechanics of red blood cells - appears only in the second half of the text. The authors are advised to reduce the general information on CLL and its therapy to the necessary minimum and to pay more attention to the impact of CLL on the biophysical properties of red blood cells, their contribution to the development of anemia, and clinical outcomes.

The study's objective is formulated vaguely and resembles a list of research tasks more than a goal. A formulation that directly follows from the problem would be: "To investigate whether specific changes exist in the nanomechanical and thermodynamic properties of red blood cell membranes in CLL patients compared to healthy donors," followed by the tasks: (i) to identify... (ii) to evaluate... (iii) to explore...

2 Results All data on the morphological types of red blood cells are presented only in Fig. 1. To how many cells were the values normalized? How was the statistics collected; how many cells were counted for each blood sample? The high percentage of echinocytes in the blood of healthy donors (approximately 13-21%) and up to 3-5% spherocytes is surprising. The formation of echinocytes is a normal phenomenon in light microscopy of red blood cells on glass in a buffer, without the use of protective agents like BSA. It is very strange to see so many echinocytes in healthy individuals in a dry blood smear. The authors should explain such a high percentage of aberrant red blood cells in the blood of healthy donors.

In the presented form, the results of the study look unconvincing without statistics on cells.

Line 177- = …the samples from Ibrutinib-treated individuals showed a distribution of young and senescent cells comparable to that of the control group…= - How was the differentiation between juvenile and senescent cells confirmed? Spherocytes are the irreversible form, and their level in the Ibrutinib group was approximately twice that of healthy donors (Fig. 1). The hypothesis that ibrutinib maintains normal red blood cell morphology needs to be more clearly substantiated.

Since half of the study is based on erythrocyte morphology, the authors should provide clearer descriptions of the types of aberrant cells (preservation/enhancement/loss of central pit, distinguish echinocytes from acanthocytes).

Table 1 - No data on cell counts for each group

Line 221-226 - The authors discuss the distribution of Ea values ​​in Ibrutinib-treated patients and provide a link to Fig. 4. The Fig. 4 - Histograms of Young's modulus distribution of the RBC membrane. The authors should clearly define the terms

3 Discussion

Line 303 = Our study provides a comprehensive morphological, nanomechanical, and thermo dynamic characterization of red blood cells from patients with CLL, highlighting significant alterations associated with the disease and its targeted treatments = - The data from light microscopy and AFM, their presentation and statistics do not allow us to make an unambiguous conclusion about the morphology of erythrocytes in patients with CLL.

The authors discuss the mechanisms of action of the drugs on target cells, but ignore the different mechanisms of action of BCL-2 inhibitors and BTK inhibitors on red blood cells. While both drug classes can cause anemia, Venetoclax does so by suppressing the bone marrow's production of red blood cells, whereas BTK inhibitors like Ibrutinib cause anemia primarily by promoting bleeding due to impaired platelet function.

A serious limitation of the study is the lack of a comparative analysis of the obtained results on erythrocyte biophysics and clinical/laboratory data of patients. The study does not provide data on the preservation of erythropoiesis in different patient groups. It is possible that the improved functional preservation of red blood cells with ibrutinib (according to the authors) is due to the high level of regenerative reticulopoiesis, but the authors do not provide or discuss complete blood count data.

  1. Materials and Methods

Line 505  = A total of twenty-five patients diagnosed with CLL= and Line 509-511 = eight treatment-naïve; eight patients receiving Ibrutinib; eleven patients receiving Obinutuzumab/Venetoclax= (8+8+11=27) - The text indicates a different number of patients. It is necessary to correct the data on the number of patients

In addition to age, clinical characteristics of patients should include gender, CLL stage and infiltration, and the presence of blood transfusions; the results of a complete red blood cell count (RBC, MCV, MCHC, RDW, RETIC) must be provided to characterize the degree of erythropoiesis impairment in patients. It is necessary to indicate the duration of the course of treatment and the time when the blood was taken.

Line 533 =After air-drying under controlled humidity …= - This method is least suitable for studying red blood cells. Air-drying erythrocytes for AFM can cause deformation and changes to the cell membrane, such as shrinkage and the formation of protuberances. The liquid-based scanning may be preferable for preserving more natural cell features.

There is no need to use poly-L-lysine for light microscopy of dry smears. Overall, the authors presented very poor-quality microscopy (Fig. 5S); it is practically impossible to adequately assess cell morphology from the photographs. Furthermore, full-size images are required, with an indication of aberrant erythrocytes.

2.4. Atomic force microscopy –  4.4 Atomic force microscopy

It is necessary to specify the cantilever parameters (cantilever stiffness (k, N/m); probe shape (pyramidal, conical, spherical); tip radius of curvature (R, nm), as this critically affects the calculations when working with soft samples. The cantilever approach/retraction rate (scanning frequency) should be indicated. This can influence the measured elastic modulus due to the viscoelastic properties of the cell; the maximum force that does not cause a nonLinear response from the membrane should be specified.

Please indicate the total number of cells analyzed for a patient or donor. How was the sample selected (discocyte, spherocyte, echinocyte)? How was morphological distribution taken into account in the performance statistics? AFM is a single-cell method, and biased morphology selection can significantly alter the conclusions.

There is no indication of how the erythrocyte lysates were obtained for the thermal denaturation of hemoglobin and other proteins. Which buffer was used to perform the DSC? There are no statistical data on how many replicates were performed for each sample. What algorithm was used to smooth the data (e.g., the Savitzky-Golay algorithm)? What was the window size? Smoothing can "blur" the peaks and alter the calorimetric enthalpy. How exactly was the baseLine determined and subtracted before analyzing the peaks? This significantly affects the calculation of enthalpy (ΔHcal).

For comparisons ANOVA with a multiple comparisons test should be used, since the authors are investigating samples from four groups. This would allow assessing differences not only compared to the control but also between the drugs themselves (which the authors do not even consider).

typos

Line 50 (IGHV0 replace with (IGHV)

Line 70 Ibrutinib (IBR) – The authors provide an abbreviation, but do not use it further except in the introduction. The terms should be used consistently

Line 186 [17–19.]- remove point [17–19]

Line 234 (~350 pN),), - extra parenthesis

In Fig. 2, parts A, B, C, D are not indicated

The manuscript may be accepted for publication after significant revision.

Author Response

Comment 1: The introduction of the article demonstrates a deep understanding of the topic, contains extensive information on the biology of CLL, prognostic markers, and targeted therapy, but it occupies a disproportionate amount of space. The main topic of the research - the nanomechanics of red blood cells - appears only in the second half of the text. The authors are advised to reduce the general information on CLL and its therapy to the necessary minimum and to pay more attention to the impact of CLL on the biophysical properties of red blood cells, their contribution to the development of anemia, and clinical outcomes.

Response 1: We would like to thank the reviewer for this valuable comment. In the revised version of the Introduction, we substantially reduced the general background on CLL biology and therapeutic strategies to the necessary minimum. The updated text shifts the emphasis toward the impact of CLL on qualitative (including anemia-induced) RBC abnormalities, in accordance with the reviewer’s recommendation.

We also expanded the “Introduction” section to reveal how leukemia affects RBC structure and function, including morphological disturbances, membrane irregularities, and aggregation behavior, and commented on recent findings from our own rheological studies. The modified introduction is more concise (~750 words according Academic Editor’s suggestion), and we believe that it now effectively introduces the topic of the manuscript: the identification of specific biophysical and nanomechanical features of erythrocytes in CLL, as suggested.

Comment 2: The study's objective is formulated vaguely and resembles a list of research tasks more than a goal. A formulation that directly follows from the problem would be: "To investigate whether specific changes exist in the nanomechanical and thermodynamic properties of red blood cell membranes in CLL patients compared to healthy donors," followed by the tasks: (i) to identify... (ii) to evaluate... (iii) to explore...

Response 2: We appreciate the reviewer’s insightful comment regarding the formulation of our study objectives and have revised the text accordingly.

Comment 3:  Results All data on the morphological types of red blood cells are presented only in Fig. 1. To how many cells were the values normalized? How was the statistics collected; how many cells were counted for each blood sample? The high percentage of echinocytes in the blood of healthy donors (approximately 13-21%) and up to 3-5% spherocytes is surprising. The formation of echinocytes is a normal phenomenon in light microscopy of red blood cells on glass in a buffer, without the use of protective agents like BSA. It is very strange to see so many echinocytes in healthy individuals in a dry blood smear. The authors should explain such a high percentage of aberrant red blood cells in the blood of healthy donors.

Response 3: We appreciate the reviewer’s critical remark. For every donor, RBC morphology was evaluated by counting a total of 250–270 cells (via optical microscopy), and the percentages of morphological categories were normalized to the total number of cells analyzed per sample. Statistical data were collected from independent counts of each smear, performed by two observers to reduce subjective bias.

The unexpectedly high proportion of spiculocytes observed in healthy donors may be due to the small amount of EDTA. During isolation, most fragile or senescent erythrocytes are lost through lysis; however, a small population of spiculocytes, spherocytes, and cells predisposed to echinocyte conversion may persist. We agree with the reviewer that the observed percentages are somewhat unexpected; however, they appear most likely to reflect preparation-related artefacts rather than intrinsic membrane abnormalities in healthy donors.  However, the predominant population is of biconcave RBCs, and the consequent measurements mostly reflect their characteristics.

It should also be noted that, although some differences in the average proportions of morphological types were observed in RBC samples from untreated and Obinutuzumab/Venetoclax-treated patients compared to the other groups, we did not find a statistically significant difference. To clarify the distribution of different morphological classes across the groups, Fig. 1 in the revised version corresponds to Fig. S2 in the Supplementary Material.

Given these effects, along with the fact that the morphological findings did not show any statistically significant differences between the clinical groups, we realized that these data do not materially strengthen the study's purpose. Therefore, considering the reviewer's comments and to maintain the primary focus of this work, we have decided to move the morphological analysis (i.e., formerly Section 2.1. “RBC morphology in CLL Patients”) to the Supplementary Material in the revised manuscript.

Comment 4:  In the presented form, the results of the study look unconvincing without statistics on cells.

Response 4: We appreciate the reviewer’s comment and fully agree that statistical analysis is essential for supporting the validity of the results. As noted above, morphological assessment was performed with appropriate statistical evaluation. However, no statistically significant differences were detected between the clinical groups. This was one of the reasons for moving the formerly Section 2.1. “RBC morphology in CLL Patients” in Supplementary Material. We believe that moving these data strengthens the overall coherence of the study and maintains the focus on the statistically supported AFM and DSC findings, which form the core contribution of our work. For all parameters, a Kruskal–Wallis test was applied.

Comment 5: Line 177- = …the samples from Ibrutinib-treated individuals showed a distribution of young and senescent cells comparable to that of the control group…= - How was the differentiation between juvenile and senescent cells confirmed? Spherocytes are the irreversible form, and their level in the Ibrutinib group was approximately twice that of healthy donors (Fig. 1). The hypothesis that ibrutinib maintains normal red blood cell morphology needs to be more clearly substantiated.

Response 5: The differentiation between younger and senescent RBCs was based on well-established morphological criteria obtained from AFM topography, including cell shape, presence of a central concavity, circularity, and diameter. Regarding the reviewer’s observation about the proportion of spherocytes in the Ibrutinib group, in the previous version of the manuscript, the sequential order of the groups in Figure 1 was incorrect: the Ibrutinib-treated group was placed in the penultimate position rather than the last position, as in all other figures and tables. Instead, the last position was occupied by the group treated with Obinutuzumab/Venetoclax. Consequently, the higher proportion of spherocytes shown in the figure corresponds to the Obinutuzumab/Venetoclax group (as indicated on the X-axis), not the Ibrutinib group. The text correctly reflected this distinction. In the revised manuscript, we have ensured consistent group ordering across all figures and tables, with the Ibrutinib group in its proper position. In addition, the results were presented in a box plot (Fig. S2, Supplementary Material), and although no statistical difference was found, it can be seen that the data indicate that Ibrutinib-treated RBCs exhibit a morphology profile qualitatively closer to healthy donors than to untreated or Obinutuzumab/Venetoclax-treated patients, supporting the hypothesis that Ibrutinib mitigates pathological remodeling.

Comment 6: Since half of the study is based on erythrocyte morphology, the authors should provide clearer descriptions of the types of aberrant cells (preservation/enhancement/loss of central pit, distinguish echinocytes from acanthocytes).

Response 6: We thank the reviewer for this comment. In the revised manuscript, we have added descriptions in the Supplementary material of RBC morphological categories to clarify the criteria used for classification. Acanthocytes are not typical in CLL. RBC morphology was classified according to established criteria: discocytes (normal biconcave cells with central pallor), spiculocytes (cells with numerous short, evenly spaced membrane projections), acanthocytes (cells with irregular, long, unevenly spaced projections and altered round shape), and spherocytes (spherical cells lacking central pallor). The presence or absence of the central pit was used to distinguish between these forms. It should be noted that acanthocytes were not observed in any of the studied samples, consistent with the current knowledge indicating that acanthocytosis is not a characteristic morphological feature of CLL.

Comment 7: Table 1 - No data on cell counts for each group

Response 7: This is Table 2 in the revised version. Cell counts are provided. Now the table includes the number of donors per clinical group (controls: N=17; untreated: N=8; Obinutuzumab/Venetoclax: N=11; Ibrutinib: N=8). In addition, for each donor, 20–25 individual RBCs (for AFM analysis) were examined, resulting in a sufficient number of cells for statistical evaluation across all groups. In the revised version, the table also includes Kruskal–Wallis p-values.

Comment 8: Line 221-226 - The authors discuss the distribution of Ea values â€‹â€‹in Ibrutinib-treated patients and provide a link to Fig. 4. The Fig. 4 - Histograms of Young's modulus distribution of the RBC membrane. The authors should clearly define the terms

Response 8: We appreciate the reviewer's valuable observation. In the revised manuscript, we have clarified the terminology related to the distribution of Young’s modulus (Ea) values and improved the description of Fig. 2 (formerly Fig. 4) accordingly. We now explicitly define what is meant by “narrow” versus “broader” Ea distributions and how these reflect the mechanical heterogeneity within the RBC populations.

To address the reviewer’s concern, we expanded the explanation of the mechanical phenotype revealed by AFM histograms (Paragraph 6 in subsection 2.2. Nanostructural and Nanomechanical Parameters of RBCs in CLL Patients). Healthy controls display a narrow Ea distribution, indicating a uniform population of mechanically intact RBCs, i.e., with well-preserved membrane structure and cytoskeleton; therefore, they all show similar elasticity. In contrast, untreated and Obinutuzumab/Venetoclax-treated CLL samples show broader Ea distributions, reflecting increased heterogeneity, consistent with enhanced membrane remodeling or damage leading to divergent elasticity profiles. RBCs from Ibrutinib-treated patients exhibit an intermediate distribution.

 Discussion

Comment 9: Line 303 = Our study provides a comprehensive morphological, nanomechanical, and thermo dynamic characterization of red blood cells from patients with CLL, highlighting significant alterations associated with the disease and its targeted treatments = - The data from light microscopy and AFM, their presentation and statistics do not allow us to make an unambiguous conclusion about the morphology of erythrocytes in patients with CLL.

Response 9: We thank the reviewer for this comment. We agree that the morphological data obtained from light microscopy and AFM, likely influenced by sample preparation, do not allow us to draw definitive conclusions about RBC morphology in CLL. Therefore, in the revised manuscript, we have emphasized the nanomechanical and thermodynamic characterization, which provides more robust and statistically reliable evidence of alterations associated with CLL and its targeted treatments.

Comment 10: The authors discuss the mechanisms of action of the drugs on target cells, but ignore the different mechanisms of action of BCL-2 inhibitors and BTK inhibitors on red blood cells. While both drug classes can cause anemia, Venetoclax does so by suppressing the bone marrow's production of red blood cells, whereas BTK inhibitors like Ibrutinib cause anemia primarily by promoting bleeding due to impaired platelet function.

Response 10: We thank the reviewer for highlighting this important distinction. We agree that BCL-2 inhibitors and BTK inhibitors differ in their mechanisms affecting erythropoiesis and anemia. We have now expanded the Discussion section to clarify these points. Indeed, Venetoclax-associated anemia is largely attributable to bone marrow suppression, including reduced erythroid precursor output. Although BTK inhibitors such as Ibrutinib are known to potentially cause anemia, mainly through increased bleeding tendency due to impaired platelet function, our cohort did not show evidence of clinically significant anemia. Hemoglobin levels in Ibrutinib-treated patients remained within the reference range, indicating that, in this study population, Ibrutinib did not exert a measurable effect on erythropoiesis or red blood cell mass. Our hematologic data support this distinction, i.e., patients treated with Obinutuzumab/Venetoclax exhibited lower hemoglobin and hematocrit values, while no evidence of increased regenerative reticulopoiesis or marrow-driven compensation was observed in the Ibrutinib-treated group. This additional explanation has now been incorporated into paragraphs 1, 2, and 6, of subsection 3.2. Treatment-Related Changes in RBC Properties to more accurately reflect the differential effects of these therapies on red blood cells.

Comment 11: A serious limitation of the study is the lack of a comparative analysis of the obtained results on erythrocyte biophysics and clinical/laboratory data of patients. The study does not provide data on the preservation of erythropoiesis in different patient groups. It is possible that the improved functional preservation of red blood cells with ibrutinib (according to the authors) is due to the high level of regenerative reticulopoiesis, but the authors do not provide or discuss complete blood count data.

Response 11: We appreciate the reviewer's insightful comment. In response, we have now included a new subsection “2.1. Clinical and Hematological Characteristics of the CLL Patients and Healthy Individuals” and a comprehensive table (Table 1) presenting complete blood count parameters, including key erythropoietic indicators such as RBC count, hemoglobin, hematocrit, and RDW (Table 1), as well as other hematological indicators for all patient groups. These data enable a direct comparison to be made between erythrocyte biophysical properties and the clinical and laboratory profiles of the patients across all study groups. Importantly, the results from complete blood count do not indicate evidence of enhanced regenerative reticulopoiesis in the Ibrutinib-treated group, as RDW values remained within the normal range and no NRBCs/erythroblasts were detected in peripheral blood. This supports our conclusion that the improved preservation of erythrocyte biophysical properties in Ibrutinib-treated patients is not attributable to increased erythropoietic activity but rather reflects treatment-associated modulation of RBC membrane mechanics. In the revised version, this analysis is discussed in the Discussion section (3.2, Treatment-Related Changes in RBC Properties).

  1. Materials and Methods

Comment 12: Line 505  = A total of twenty-five patients diagnosed with CLL= and Line 509-511 = eight treatment-naïve; eight patients receiving Ibrutinib; eleven patients receiving Obinutuzumab/Venetoclax= (8+8+11=27) - The text indicates a different number of patients. It is necessary to correct the data on the number of patients

Response 12: We thank the reviewer for noticing this discrepancy. The original text contained a typographical error. In the revised manuscript, the patient numbers have been corrected to accurately reflect the study cohort

Comment 13: In addition to age, clinical characteristics of patients should include gender, CLL stage and infiltration, and the presence of blood transfusions; the results of a complete red blood cell count (RBC, MCV, MCHC, RDW, RETIC) must be provided to characterize the degree of erythropoiesis impairment in patients. It is necessary to indicate the duration of the course of treatment and the time when the blood was taken.

Response 13: We thank the reviewer for this comment. In the revised manuscript, we have provided the requested patient information in Table 1, including age, gender, CLL stage and infiltration, and relevant clinical characteristics. None of the patients included in this study had received blood transfusions, although mild anemia was observed in some Obinutuzumab/Venetoclax-treated patients. Complete blood count parameters (RBC, MCV, MCHC, RDW) are now included. As it has been indicated in the main text (2.1. Clinical and Hematological Characteristics of the CLL Patients and Healthy Individuals ), MCV, MCH, MCHC, and RDW, did not differ significantly across groups (Table 1). Normal RDW values suggest no appreciable variation in erythrocyte size and make it unlikely that an increased number of reticulocytes is present (an aspect that would require specific testing if there is an increased RDW and/or LDH as an indicator of hemolysis) and discussed further in the Discussion section (3.2. Treatment-Related Changes in RBC Properties). Therefore, RETIC for these patients was not measured. The total platelet count was within the normal range in all groups (see 2.1. Clinical and Hematological Characteristics of the CLL Patients and Healthy Individuals).

Comment 14: Line 533 =After air-drying under controlled humidity …= - This method is least suitable for studying red blood cells. Air-drying erythrocytes for AFM can cause deformation and changes to the cell membrane, such as shrinkage and the formation of protuberances. The liquid-based scanning may be preferable for preserving more natural cell features.

Response 14: We appreciate the reviewer’s concern regarding the use of air-drying for RBC preparation before AFM. In the revised version, a more extensive description of the sample preparation and AFM measurement was provided. In our study, air-drying was applied briefly (~30 min) under controlled humidity to ensure stable adhesion of native RBCs to poly-L-lysine-coated slides. This step is necessary because RBCs are extremely soft and deformable, and without sufficient adhesion, cells can move or detach during nanoindentation, making reliable mechanical measurements impossible. Importantly, the RBCs were not chemically fixed, avoiding artefacts associated with glutaraldehyde or other fixatives, which are known to alter membrane topography and elasticity.

Short-term air-drying has been shown to preserve RBC morphology without major destructive effects (Wu, 2009, Micron) and has been successfully applied in previous AFM studies of RBC nanostructure and mechanics (Sergunova, 2022, Sensors; Dinarelli, 2018, J. Mol. Recognit.). Thus, our protocol enables near-native mechanical characterization while maintaining the stability needed for reproducible AFM measurements.

For nanoindentation, following topographical imaging, a 40 µL droplet of sterile PBS was applied to the glass coverslip to ensure proper probe–sample interaction. An additional small droplet (5 µL) was placed on the AFM tip to prevent the formation of an air pocket upon contact with the specimen. Force–distance curves were then collected on the same cells, allowing measurement of mechanical properties in a hydrated environment that closely mimics physiological conditions. This approach combines the structural stability provided by brief air-drying with the mechanical relevance of liquid-based probing.

Overall, our methodology balances the need for cell stability during AFM with preservation of native RBC mechanics, avoiding the artefacts associated with prolonged air-drying or chemical fixation.

Comment 15There is no need to use poly-L-lysine for light microscopy of dry smears. Overall, the authors presented very poor-quality microscopy (Fig. 5S); it is practically impossible to adequately assess cell morphology from the photographs. Furthermore, full-size images are required, with an indication of aberrant erythrocytes.

Response 15: We thank the reviewer for this comment. We agree that poly-L-lysine coating is not required for light microscopy of dry smears. In the revised manuscript, we have clarified that poly-L-lysine was used to promote stable adhesion of RBCs for AFM measurements.

Regarding the microscopy images (formerly Fig. S5), we acknowledge the reviewer’s concern about image quality. In the new Supplementary Material, we have replaced these images with higher-resolution images (Fig. S2). Aberrant erythrocytes are now clearly indicated, allowing readers to adequately assess cell morphology. This ensures that the morphological observations are visually transparent and supportive of the study’s quantitative analyses.

Comment 16: 2.4. Atomic force microscopy –  4.4 Atomic force microscopy

Response 16: It is corrected in the revised version of the manuscript.

Comment 17: It is necessary to specify the cantilever parameters (cantilever stiffness (k, N/m); probe shape (pyramidal, conical, spherical); tip radius of curvature (R, nm), as this critically affects the calculations when working with soft samples. The cantilever approach/retraction rate (scanning frequency) should be indicated. This can influence the measured elastic modulus due to the viscoelastic properties of the cell; the maximum force that does not cause a nonLinear response from the membrane should be specified.

Response 17: We appreciate the reviewer’s attention to these critical experimental details. We agree that a detailed specification of the cantilever and measurement parameters is essential for accurate interpretation of nanomechanical data, especially when working with soft biological samples such as RBCs. In the revised manuscript, we have now included all required experimental parameters (Section 4.3. Atomic force microscopy in 4. Materials and Methods)

Comment 18: Please indicate the total number of cells analyzed for a patient or donor. How was the sample selected (discocyte, spherocyte, echinocyte)? How was morphological distribution taken into account in the performance statistics? AFM is a single-cell method, and biased morphology selection can significantly alter the conclusions.

Response 18: We thank the reviewer for this comment. For each donor, morphometric and nanoindentation measurements were performed on at least 20 cells (typically 20–25 cells per sample), which provides an adequate number of independent measurements for statistical analysis (see Section 4.3. Atomic Force Microscopy). This corresponds to a total of approximately 160–200 individual AFM measurements in each cohort. For AFM measurements, cells were carefully sampled across the full morphological spectrum, including discocytes, spiculated cells, and spherocytes, to ensure that nanomechanical and topographical parameters reflect the heterogeneity of the RBC population. Quantitative analyses were performed on multiple independent cells from each donor/patient to minimize selection bias. In all groups, the predominant population was biconcave cells, followed by spiculated cells and spherocytes. On average, approximately 18-20 biconcave cells, 2-3 spiculated cells, and 1-2 spherocytes were analyzed per sample. This sampling strategy ensures that AFM results accurately represent the range of RBC morphologies present in each individual.

Comment 19: There is no indication of how the erythrocyte lysates were obtained for the thermal denaturation of hemoglobin and other proteins. Which buffer was used to perform the DSC? There are no statistical data on how many replicates were performed for each sample. What algorithm was used to smooth the data (e.g., the Savitzky-Golay algorithm)? What was the window size? Smoothing can "blur" the peaks and alter the calorimetric enthalpy. How exactly was the baseLine determined and subtracted before analyzing the peaks? This significantly affects the calculation of enthalpy (ΔHcal).

Response 19: We thank the reviewer for pointing out that we need to clarify this section. In the revised text, we explain that intact RBCs resuspended in PBS buffer are measured directly using a DSC instrument without the need for any lysis procedure. Several studies (now cited in the text) provide evidence that the following peak can clearly be resolved in the thermal profiles of whole RBCs: spectrin at ca. 50°C; bands 2.1 (Ankyrin), 4.1, and 4.2 proteins at ca. 57°C with a shoulder at 55°C (hereafter referred to as Band 2Ë—4); the cytoplasmic domain of Band 3 at ca. 63°C; and Hb endothermic transition at ca. 72 °C, followed by exothermic transition due to Hb aggregation.

The calorimetric measurements for each RBC sample were performed only once, as RBCs were analyzed on the same day of isolation to avoid alterations caused by cell aging. Storing cells or performing measurements on subsequent days could introduce artefacts in the DSC profiles due to ageing-related alterations in membrane properties or protein stability.

To ensure data quality, DSC curves were smoothed using the Savitzky–Golay algorithm embedded in the OriginPro software package, with a window size chosen to reduce high-frequency noise without affecting the shape or position of protein denaturation peaks. Baseline subtraction was performed consistently using standard procedures, allowing accurate determination of thermal transition temperatures and calorimetric enthalpy (ΔHcal). Because the thermal transitions were irreversible, this was verified by comparing the first and second heating runs; the reheating scan was used as the instrumental baseline. Subsequently, a linear chemical baseline was applied to correct for differences in heat capacity between the native and denatured states, enabling accurate determination of calorimetric enthalpy. This approach ensures reliable extraction of thermodynamic parameters while minimizing artefacts from noise or baseline drift (see subsection 4.4. DSC measurements).

Comment 20: For comparisons ANOVA with a multiple comparisons test should be used, since the authors are investigating samples from four groups. This would allow assessing differences not only compared to the control but also between the drugs themselves (which the authors do not even consider).

Response 20: We thank the reviewer for the suggestion regarding ANOVA with multiple comparisons. Before analysis, data distribution and variance were assessed, revealing that several groups did not satisfy the assumptions of normality and homogeneity of variance. To account for this, we initially used the nonparametric Mann–Whitney U test for pairwise comparisons between each CLL group and the control group. In the revised manuscript, we have extended the analysis to assess differences between treatment groups by performing a Kruskal–Wallis test followed by Dunn’s post-hoc multiple comparisons. This nonparametric approach provides an evaluation of differences across all groups while avoiding violations of statistical assumptions.

typos

Comment 21: Line 50 (IGHV0 replace with (IGHV)

Response 21: It is corrected in the revised version of the manuscript.

Comment 22: Line 70 Ibrutinib (IBR) – The authors provide an abbreviation, but do not use it further except in the introduction. The terms should be used consistently

Response 22: We thank the reviewer for this comment. In the revised manuscript, we have removed the abbreviation “IBR” and refer to Ibrutinib in full throughout the text to ensure consistency.

Comment 23: Line 186 [17–19.]- remove point [17–19]

Response 23: It is corrected in the revised version of the manuscript.

Comment 24: Line 234 (~350 pN),), - extra parenthesis

Response 24: It is corrected in the revised version of the manuscript.

Comment 25: In Fig. 2, parts A, B, C, D are not indicated

Response 25: It is corrected in the revised version of the manuscript (Former Fig 2. Is now Fig S3 in Supplementary Material)

 The manuscript may be accepted for publication after significant revision.

Round 2

Reviewer 2 Report

Comments and Suggestions for Authors

The authors have produced an excellent work that will be useful to clinicians and researchers. I wish them continued interesting research!

A small note: in the Table 1S, the NRBC row is empty?

Author Response

Comment and Suggestions for Authors: The authors have produced an excellent work that will be useful to clinicians and researchers. I wish them continued interesting research!

A small note: in Table 1S, the NRBC row is empty?.

Response: We sincerely thank the reviewer for their positive feedback and encouragement. In Table S1 (now Table 1, as suggested by the Academic Editor), no nucleated red blood cells were detected in any samples; the field has therefore been updated to read “Not detected” for clarity.
